# Effects of Psychotherapy on the Problem Behaviors of Humidifier Disinfectant Survivors: The Role of Individual Characteristics and Adaptive Functioning

**DOI:** 10.3390/healthcare11152179

**Published:** 2023-08-01

**Authors:** Min Joo Lee, Yubin Chung, Soeun Hong, Hun-Ju Lee, Gippeum Park, Sang Min Lee

**Affiliations:** 1Department of Education, Korea University, Seoul 02841, Republic of Korea; mjbravo@korea.ac.kr (M.J.L.); yubin90@gmail.com (Y.C.); soeunhong@gmail.com (S.H.); joy3151@naver.com (G.P.); 2University Industry Foundation, Yonsei University, Seoul 03722, Republic of Korea; hunjulee@yonsei.ac.kr

**Keywords:** adaptive function, effects of psychotherapy, humidifier disinfectant survivors, problem behaviors, psychological symptoms

## Abstract

This study aimed to examine group differences in the survivors of humidifier damage and the effect of individual psychotherapy on the psychological symptoms of the survivor groups, using the single group pre–post study design. A series of Wilcoxon–Mann–Whitney tests were conducted to investigate the level of psychological problems before and after psychotherapy, as well as the main and interaction effects of demographic characteristics and adaptive functioning on the treatment effects in 69 humidifier disinfectant survivors. The results demonstrated significant differences in problems with socioeconomic status (SES), life functioning, friendships, family relationships, and job adjustment in the survivor groups. Groups with high SES, low life functioning, and poor friend relationships had more problem behaviors than other groups. Problem behaviors related to friendship levels were different before and after psychotherapy. After psychotherapy, individuals with limited social connections exhibited a greater decrease in problem behaviors compared to those with strong friendships. This paper extends the international literature on the long-term consequences of environmental health hazards and the importance of tailored mental health interventions.

## 1. Introduction

It is necessary to look at not only the physical but also the psychological damage of the humidifier disinfectant survivors. This can be an important foundation for developing and organizing psychological interventions based on their psychological problems. To do this, it is necessary to examine the damage caused by humidifier disinfectants in South Korea. It was proposed that humidifier disinfectants, which have been sold since 1994, could prevent diseases caused by bacteria. People believed that this messaging, promoted by a large company, was true and that humidifier disinfectant would help prevent diseases and protect their health and that of their family members [1]. Families with newborn children also became attracted to the fact that they could effectively eliminate bacterial diseases.

In April 2011, the Asan Medical Center in Seoul reported the occurrence of a strange disease in six mothers, the symptoms of which were respiratory failure and pulmonary fibrosis [2]. These cases were distributed throughout all areas in South Korea and were not concentrated in a particular region. The Korea Centers for Disease Control and Prevention (K-CDC) conducted an epidemiological investigation into the reasons behind the occurrence of these symptoms. The K-CDC found that humidifier disinfectant, biocide water placed in humidifiers along with water, was associated with the particular symptoms being displayed by the six women.

In December 2012, a Lung Injury Investigation Committee (LIIC) was formed to investigate the damage caused by these humidifiers, and in July 2013, investigations on humidifier disinfectants and their damage status began [3]. In-depth investigations have found that the chemical products used in humidifiers are associated with a wide range of lung injuries, including interstitial pneumonitis [4,5,6]. The chemicals contained in humidifier disinfectants include substances that are fatal to the human body, such as chloromethylisothiazolinone (CMIT), methylisothiazolinone (MIT), hexamethylene guanidine phosphate (PHMG), and oligo 2-(2-ethoxy) ethoxy ethyl guanidine chloride (PGH). In particular, products containing CMIT and MIT were sold from 1994 to 2011, and approximately 9.98 million units of humidifier disinfectants were sold [7]. It is inferred that humidifier disinfectant products may have affected a random selection of people over an extended period of time [8]. Toxic chemicals were absorbed into the body through the nose, mouth, and skin via nebulizers and began to penetrate the lungs, causing irreversible damage to lung cells and widespread health damage [6,9,10,11]. As reported in a study conducted on 94 adults in Gwangmyeong City in Gyeonggi Province, 37.2% used a humidifier, and 18.1% used a humidifier disinfectant [12]. In an epidemiological survey of 1144 pregnant women, the rate of humidifier use was 28.2% [13]. Yoon et al. [8] argued that 75.6% of children in Korea used humidifiers, and 31.1% of children were exposed to humidifier disinfectants. Considering that humidifier disinfectant was also used in families with children and/or newborns, it can be inferred that the amount of damage to survivors and their families will increase. However, it is difficult to accurately estimate the number of victims [14]. As of April 2022, the number of people who applied for damages was 7685, including 1751 deaths.

Additionally, the problems of humidifier disinfectant survivors appeared not only as physical problems but also as psychological problems. Yoo et al. [15] analyzed the psychological problems faced by 26 survivors and 92 survivors’ families (45 bereaved). They found that survivors still experience anxiety and fear, even after the event has passed, and bereaved families often display alcohol abuse and insomnia. According to Leem et al. [16], 57.5% of humidifier disinfectant survivors reported depression, 55.1% reported guilt and self-blame, 54.3% reported anxiety, 27.6% reported suicidal thoughts, and 11% reported suicide attempts. Ko et al. [17] compared the mental health status of humidifier disinfectant survivors from 2018 to 2021 in a general population of 228, and the survivors reported more psychological problems, such as anxiety, depression, atrophy, and thinking problems, compared to the general population.

In 2017, the Ministry of Environment created an organization to provide psychological psychotherapy interventions for humidifier disinfectant survivors and their families. The organization, named Health Monitoring of Humidifier Disinfectant Victims (Mental Health), provides various psychological interventions, such as group psychotherapy and social improvement programs; individual psychotherapy is one of the most essential interventions that can be provided to the survivors and their families. Therefore, it is necessary to examine the effectiveness of individual psychotherapy for humidifier disinfectant survivors to alleviate their psychological distress. Several studies have shown that psychological interventions are effective for people suffering from various psychological problems, such as PTSD, stress, anger, and depression [18,19]. Howard et al. [20] argued that psychotherapy was more significant than spontaneous recovery, based on the results of a meta-analysis of the effects of psychotherapy on 2400 clients over a 30 year study period. On the other hand, while Cuijpers et al. [21] did not deny the effectiveness of psychotherapy, they reported that the effectiveness of psychotherapy was somewhat exaggerated.

The purpose of this study is to examine the effectiveness of individual therapy on psychological symptoms. Additionally, this study examines the moderating role of demographic characteristics in treatment effect to identify the characteristics of survivors who can most benefit from individual psychotherapy. By analyzing the psychological symptoms of humidifier disinfectant survivors and the effectiveness of psychotherapy on them, this study can contribute to the understanding of what constitutes an appropriate psychological intervention for humidifier disinfectant survivors.

The following are the specific research questions for this study: First, what characteristics did survivors who had significant psychological problems describe? We specifically expect that there could be statistical differences in the extent of psychological symptoms by SES, according to the findings of the Ko et al. [17] study. Second, is individual psychotherapy effective for survivors of humidifier disinfectants? According to the results of previous research [18,19,20,21], we assume that individual psychotherapy has a considerable impact in relieving psychological symptoms. Third, which survivor groups benefited the most from psychotherapy? We predict that several traits of survivors (such as the quality of their friendships) will moderate the effectiveness of individual psychotherapy in reducing psychiatric symptoms. Several scholars [22,23] note that the effectiveness of psychotherapy is a complex interaction of several traits and that what works best for one person may not work the same way for another. Tailoring treatment to the individual’s unique characteristics and needs is essential for achieving the best possible outcomes.

## 2. Materials and Methods

### 2.1. Participants

The ethical approval for this study was obtained from the Institutional Review Board of Yonsei University (No. 7001988-202104-HR-1178-02l) in South Korea. This study used data from a total of 69 individuals who survived exposure to toxic humidifier disinfectants and sought individual psychotherapy through a government support program administrated by the National Institute of Environmental Research (NIER) in 2021. Among the survivors aged 13 years and older, a total of 224 individuals received psychotherapy. However, only 69 of them voluntarily participated in this study with informed consent. The response rate among the participants was 30.8%. The age range of the survivors was 13 to 60 years (M = 38.13, SD = 14.75), and 55.1% (*n* = 38) were female. The survivors received treatment at one of the psychotherapy centers officially approved by the government program, which stipulates the conditions for a psychotherapy center to be eligible for the support program. The centers had certified therapists from the Korean Counseling Psychological Association (KCPA) or the Korean Counseling Association (KCA). The participants completed the survey of Achenbach System of Empirically Based Assessment (ASEBA) provided by HUNO Inc. (ASEBA Provider Company in Seoul, Republic of Korea), before and after treatment.

### 2.2. Treatment

Therapists who held level 2 or higher psychotherapy-related certificates issued by the KCPA or the KCA provided individual psychotherapy to survivors. One therapist conducted psychotherapy with only one or two clients. The therapists’ theoretical orientations mainly included cognitive behavioral therapy, psychodynamic therapy, interpersonal psychotherapy, and integrative therapy. The default number of sessions provided was a total of 10 sessions, with each session lasting 50 min. However, sessions could be extended based on the client’s status. The average number of sessions was 15 (SD = 7.52), with a range from 4 to 41. In each session, the psychotherapy process was monitored using the Outcome Rating Scale (ORS) and the Session Rating Scale (SRS) completed by the survivors [24]. Therapists submitted weekly session evaluation reports and completed psychotherapy records and online psychological assessments. The quality of treatment was steadily managed through regular monitoring by the government support program.

### 2.3. Measures

#### 2.3.1. Problem Behavior Scale

The Adult Self-Report (ASR) [25,26] and the Youth Self-Report (YSR) [27] were used to measure the psychological problem behaviors of participants before and after treatment. Adolescents under the age of 19 responded to the YSR, and adults responded to the ASR. In the YSR and ASR, the total scores of the problem behavior scale consist of the internalizing scale, externalizing scale, thought problems, attention problems, social problems (only for YSR), and other similar problems. Anxious/depressed, withdrawn/depressed, and somatic complaints correspond to the internalizing scale, whereas aggressive behavior, rule-breaking behavior, and intrusive behavior (only for ASR) correspond to the externalizing scale. The reliability of the problem behavior scale was 0.95 for youth and 0.97 for adults, respectively. The Korean versions of the YSR and ASR were provided by HUNO Inc., and standardized T-scores were reported based on representative samples of adolescents (11–18 years old) and adults (19–60 years old) in South Korea. The higher the score, the more serious the mental health problems: T ≥ 64 indicates clinical, T = 63–60 is borderline clinical, and T < 60 is non-clinical mental health problems, meaning that participants in a clinical or borderline clinical group are more likely to exhibit emotional and behavioral problems in adapting to situations than other participants in the same age and gender groups.

#### 2.3.2. Adaptive Functioning Scale

The degree of adaptive functioning of the participants was measured using the YSR and ASR, before and after treatment. For ASR, the total score of adaptive functioning was calculated using the scales of friends, spouse/partner, family, job, and education. For the YSR, sociality and academic performance were considered to calculate the total score of adaptive functioning. The reliability of the adaptive functioning scale was 0.56 for youth and 0.70 for adults, respectively. The lower the score, the lower the adaptive level: T ≤ 36 indicates clinical, T = 37–40 is borderline clinical, and T > 40 is non-clinical. The clinical standards of the sub-scale of adaptive functioning, including friendship (friends in ASR and sociality in YSR), family, and job, are clinical (T ≤ 30), borderline clinical (T = 31–35), and non-clinical (T > 35). For example, clinical group participants in the case of friendships were less likely to adapt to having meaningful relationships with friends than non-clinical group participants.

### 2.4. Data Analysis

Data analyses were performed using IBM SPSS version 21. First, we conducted descriptive statistics analysis to provide an overview of the demographic and life functioning variables before and after treatments. Regarding missing data, we addressed it by imputing the missing values with the mode score, following the rules of ASEBA. Second, we conducted the Mann–Whitney test for two independent groups, the Kruskal–Wallis test for three or more independent groups, and the Wilcoxon signed-rank test for paired samples to assess treatment effects (i.e., whether the total problem level changed significantly before and after treatment) [28,29,30]. Although the power analysis using G*power software with an effect size of 0.40 [31] indicated that 60 participants were sufficient to run a dependent t-test and one-way ANOVA to examine the treatment effects, we chose to perform a series of nonparametric tests to compare outcomes between multiple groups in order to obtain clearer results. Lastly, for practical significance, effect sizes using Cohen’s d were calculated and interpreted as d = 0.20 (small effect), d = 0.50 (medium effect), and d = 0.80 (large effect [32]).

## 3. Results

### 3.1. Mean Differences in Problem Behavior Scores

Mann–Whitney tests and Kruskal–Wallis tests were conducted for the group differences of each variable at the pre-treatment stage to identify groups that reported more problem behaviors than the other groups. The median and interquartile of each variable are listed in Table 1. There were no significant differences between survivors with and without compensation, between self and family members, and between male and female participants. There were significant differences among age groups; the age group of 40–50 (T_Md_ = 62.50, 54.75–72.75) displayed more problem behaviors than the age group under 19 (T_Md_ = 57.00, 48.25–63.50). There were significant differences in problem behaviors regarding SES, life functioning, friendships, family relationships, and job adjustment. For example, the SES group with the highest level (T_Md_ = 63.50, 57.00–86.25) reported more problem behaviors than the middle (T_Md_ = 56.00, 49.00–62.75) and the lowest (T_Md_ = 55.00, 48.00–63.00) levels. For life functioning, the group with the lower level (T_Md_ = 63.00, 57.00–82.50) showed more problem behaviors than the group with higher levels (T_Md_ = 53.50, 47.25–61.75). In the case of friendship, the group with poor relationships (T_Md_ = 63.00, 57.00–80.75) displayed more problem behaviors than the group with good relationships (T_Md_ = 57.00, 48.00–63.00). Similarly, in family relationships, the group with poor relationships (T_Md_ = 59.00, 57.00–89.25) reported more problem behaviors than the one with good relationships (T_Md_ = 53.50, 47.00–62.25).

### 3.2. Treatment Effects and Interaction with the Demographic and Life Functioning Variables

The Wilcoxon Signed Ranks Test was conducted with problem behaviors at treatment time points (pre- and post-intervention) to determine whether there were significant treatment effects. Problem behaviors significantly decreased with time (Z = −2.955, *p* = 0.003), with medium to large effect sizes. The results of the Wilcoxon Signed Ranks Test showed a significant main effect for time, but no significant interaction effect between time and groups of variables: survivors with and without compensation, self and family member, male and female, age, SES, family, and job adjustment. Additionally, there was a significant main effect of time and an interaction effect (Z = −2.342, *p* = 0.019) on friendship. No significant interaction effects of time and family relationships, time, life functioning, or job adjustment were found.

Because there were some significant differences in problem behaviors before and after treatment according to the levels of friendship, a post hoc analysis (simple effect comparisons) was conducted. In other words, we examined whether the level of friendship influenced the treatment effects. The quality of friendship was divided into two groups: the poor relation group (T ≤ 35), which includes standard scores of clinical and borderline clinical provided by ASEBA and the good relation group (T > 35). There was no notable difference in problem behaviors in the good relation group before (T_Md_ = 57.00, 48.00–63.00) and after (T_Md_ = 55.00, 43.00–65.00) treatment. In the case of the poor relation group, problem behaviors significantly decreased after (T_Md_ = 55.50, 50.75–67.25) treatment, compared to before (T_Md_ = 63.00, 57.00–80.75) (see Figure 1). The poor relation group reported fewer problem behaviors after treatment than the good relation group.

## 4. Discussion

This study aimed to closely examine group differences in the survivors of humidifier disinfectant damage and the effect of individual psychotherapy on the survivor groups. For this purpose, the results are as follows: First, the differences in problem behaviors at the pre-treatment stage based on socio-demographic and psychological variables were examined to identify groups that reported more problem behaviors than others. There were no significant differences between survivors with and without compensation, between self and family members, between male and female participants, and between the three age groups. However, survivors from high socioeconomic status (SES) backgrounds reported more problem behaviors compared to those from low- and middle-SES backgrounds. The higher family economic status group’s psychological symptoms may be influenced by social comparison. Ko et al. [17] also reported the results that the high-SES group among humidifier disinfectant survivors experienced more psychological problems than the other groups. Humidifier disinfectant survivors who reported a high level of life functioning, friendships, family relationships, and job adjustment were less likely to report problem behaviors. The results of this study are consistent with those of several studies reporting that good functioning in life and good interpersonal relationships are closely related to a low level of psychological problems. Furthermore, many studies have found that individuals who function well in life and have positive relationships tend to experience better physical health, increased life satisfaction, higher confidence, and lower stress and anxiety levels [33]. In this way, the correlation between psychological problems and life function appears to be that they influence each other and accelerate their impact.. The previously mentioned study findings allow us to confirm the requirement for psychological intervention in a person whose life function has been negatively affected by psychological problems.

Second, the treatment effects of individual psychotherapy and interactions with demographic and life functioning variables were analyzed. The results indicated a significant main effect for time, but no significant interaction effect between time and groups of variables: survivors with and without compensation, self and family members, male and female, age, SES, life functioning, family, and job adjustment. Specifically, the effects of individual psychotherapy were more pronounced in survivors with poor friendship, as their problem behavior scores significantly decreased following treatment.

The results of this study can be explained by the study of the life adaptation of people who have achieved psychological recovery through individual psychotherapy [33]. Research on disaster survivors has shown that some survivors recover well psychologically, while others experience psychological difficulties to the point of experiencing trauma from the disaster [34,35,36,37,38,39,40]. Through psychotherapy, individuals achieve psychological recovery by gaining social support, forming a sense of solidarity, and accepting their pain [41,42,43]. Survivors can learn to cope with stress, enhance life satisfaction, reduce psychological symptoms, function effectively at work and home, and cultivate positive relationships through the psychotherapy process. Finally, it was found that psychological problems were significantly reduced after psychotherapy in the group with poor friendships. This result is in line with the research findings that posit that social support helps psychological recovery after an accident or disaster [44,45,46,47]. In addition, this is consistent with the results of studies reporting that happiness and social support are closely related to life satisfaction [48,49,50]. Mexico, which is exceptional in the correlation between the average income level, crime rate, and happiness index, has a lower income level and higher crime rate than the United States but a higher happiness index. This index can be attributed to the cultural characteristics of Mexico, where there are stronger family bonds and a higher frequency of contact among family members than in the United States [51]. Individuals with social support are less susceptible to stress and experience fewer psychological difficulties. The group with poor friendships may improve their ability to seek social support through psychotherapy, which leads to a reduction in psychological problems. In summary, this indicates that treatment may enhance the survivors’ ability to seek social support.

The limitations of this study are as follows. Firstly, the subjects were a unique group of survivors of humidifier disinfectant disasters. Therefore, it is necessary to be careful when interpreting the research results for the general public. Secondly, there is a limitation in verifying statistical significance because the number of subjects was relatively small. Although G*Power provided a minimum sample size of 60, it was small to divide the sample into adults and minors and perform all the analyses. Thus, it is necessary to replicate these results by increasing the number of subjects in the future. Thirdly, we considered demographic characteristics (e.g., age, SES, compensation presence, etc.) as moderators for treatment effectiveness. However, other potentially critical moderation variables such as the severity of the health damages by the humidifier disinfectant use, losses of participating family members, the duration of victimization, taking psychotropic drugs, and the number of psychotherapy sessions should also be included as moderation variables in future studies. Fourthly, this study focused on verifying the effectiveness of a distinctive group of social disaster survivors who immediately needed psychotherapy support. Due to the nature of these subjects, it is ethically problematic to randomly assign survivors to the control group, effectively suspending them from psychotherapy. To explore the effect of psychotherapy in this situation, where a control group could not be secured due to ethical issues, this study observed changes before and after psychotherapy through a time-series design. Because this study has limitations in that it cannot control factors for internal validity from the experimental design, attention should be paid to the interpretation of the results of the study. Lastly, all items were self-reported; therefore, the response could be biased due to faking negatively or positively (e.g., social desirability). In future studies, the researchers need to utilize objective measures such as significant others’ observations.

Based on the findings and limitations of this study, the following recommendations can be made for practice, research, and management. Tailored interventions addressing these specific concerns of survivors may be beneficial. Future studies should replicate this research with larger sample sizes to enhance statistical power and allow for subgroup analyses, such as comparing adult and minor survivors separately. In addition to demographic variables, researchers should consider including other critical factors as moderation or mediation variables, such as the severity of health damages, losses of family members, duration of victimization, and number of psychotherapy sessions, to better understand their influence on treatment effectiveness. Objective measures, such as observations from significant others, should be incorporated alongside self-reported measures to minimize response biases and enhance the validity of the research findings. Public health authorities and policymakers should consider implementing and promoting accessible mental health services for survivors of environmental disasters. Adequate resources and support should be allocated to ensure timely and effective psychological interventions are available to those in need. Managers of mental health facilities and organizations should encourage interdisciplinary collaboration and research partnerships to further explore the psychological impacts of environmental disasters. This can help develop evidence-based practices and interventions tailored to the specific needs of different survivor groups.

It is important to acknowledge the limitations of this study, including the small sample size, the specificity of the population studied, the lack of a control group, and the reliance on self-reported measures. These limitations should be taken into account when interpreting our findings, and recommendations should be considered in light of these limitations. Further research and replication studies are needed to validate the findings and address the identified limitations. Despite all the limitations, this study has meaning: (1) It examined the effect of psychotherapy on survivors of social disasters. (2) Survivors were classified according to differences in their ability to adapt to life, even though they experienced the same disaster. The differences in the effectiveness of psychotherapy according to the classified group were examined. (3) It is especially meaningful that we explored which group showed the greatest change due to psychotherapy. Overall, this study provides insights into the group differences among survivors of humidifier disinfectant damage and underscores the positive impact of individual psychotherapy on psychological recovery, particularly in individuals with poor friendship networks.

In conclusion, this study aimed to investigate group differences among the survivors of humidifier disinfectant damage and the effects of individual psychotherapy on their psychological symptoms. The findings of this study contribute to our understanding of the psychological impact of such environmental disasters from an international perspective. The results demonstrate the effectiveness of individual psychotherapy in reducing psychological symptoms among survivors of humidifier disinfectant damage. The treatment significantly decreased problem behaviors over time, with medium to large effect sizes. Significant interaction effects were found between treatment effects and friendship levels on problem behaviors. This study highlights the importance of considering the quality of friendships in psychological interventions for survivors. Individuals with poor friendships showed significant improvements in problem behaviors following psychotherapy, emphasizing the role of social support in psychological recovery after disasters or accidents. Overall, this study contributes to the scientific and civil communities’ understanding of the psychological consequences of environmental disasters, underscores the positive impact of individual psychotherapy on survivors’ psychological recovery, and highlights the importance of addressing social support and relational approaches in interventions for disaster survivors.

## 5. Conclusions

The paper aimed to investigate the psychological impact of environmental disasters on survivors and the effects of individual psychotherapy on their psychological symptoms. The study found that individual psychotherapy was effective in reducing psychological symptoms among survivors of humidifier disinfectant damage. The treatment significantly decreased problem behaviors over time, with medium to large effect sizes. This study also highlighted the importance of considering the quality of friendships in psychological interventions for survivors, as individuals with poor friendships showed significant improvements in problem behaviors following psychotherapy. The paper suggested tailoring interventions to address survivors' specific issues, replicating the study with a larger sample size, and incorporating objective measures alongside self-report measures to increase the validity of the findings. Public health authorities and policymakers should consider implementing and promoting accessible mental health services for survivors of environmental disasters, and adequate resources and support should be allocated to ensure timely and effective psychological interventions are available to those in need. Managers of mental health facilities and organizations should encourage interdisciplinary collaboration and research partnerships to further explore the psychological impacts of environmental disasters.

## Figures and Tables

**Figure 1 healthcare-11-02179-f001:**
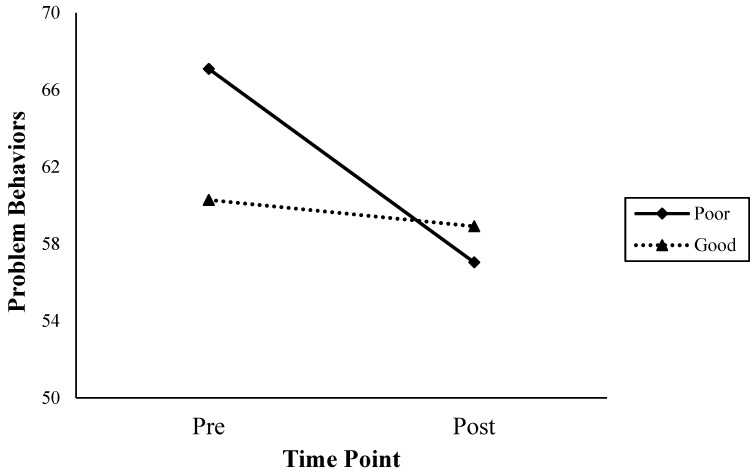
Interaction effect between quality of friendship (poor or good) and problem behaviors by time point.

**Table 1 healthcare-11-02179-t001:** Descriptive statistics of problem behaviors according to research variables at pre- and post-treatment stages.

Variable		*n*	Pre-Test		Post-Test		Cohen’*d*
			*M(SD)*	*Md(IQR)*	*M(SD)*	*Md(IQR)*	
Compensation Presence	With Compensation	26	63.77(17.138)	59.50(52.75–80.75)	57.88(12.854)	55.00(49.50–65.25)	0.39
	Without Compensation	43	61.81(13.512)	60.00(52.75–80.75)	58.53(15.906)	54.00(45.00–66.00)	0.22
Compensation Subject	Self	44	64.36(14.989)	61.50(55.00–71.25)	60.86(15.200)	58.50(49.25–67.75)	0.23
	Family member	25	59.36(14.454)	57.00(49.50–66.50)	53.76(12.950)	52.00(43.50–62.50)	0.41
Gender	Male	31	59.65(16.390)	58.00(48.00–68.00)	54.97(12.828)	52.00(44.00–65.00)	0.32
	Female	38	64.92(13.294)	60.50(56.75–69.75)	61.00(15.774)	55.50(50.75–70.50)	0.27
Age	Under 19	12	54.92(9.885)	57.00(48.50–63.50)	48.17(8.100)	45.00(42.00–53.75)	0.75
	20 to 39	15	62.47(15.793)	58.00(49.00–79.00)	57.07(13.956)	52.00(47.00–65.00)	0.36
	40 to 59	42	64.76(15.303)	62.50(54.75–72.75)	61.62(15.300)	61.50(50.75–68.50)	0.21
SES	High	38	67.50(15.866)	63.50(57.00–86.25)	61.05(16.319)	59.00(50.00–68.75)	0.40
	Middle	24	56.46(11.077)	56.00(49.00–62.75)	55.83(12.345)	54.50(45.00–62.00)	0.053
	Low	7	56.57(12.012)	55.00(48.00–63.00)	51.71(10.547)	47.00(43.00–65.00)	0.43
Life functioning	High	28	54.50(12.816)	53.50(47.25–61.75)	53.64(14.952)	50.50(42.24–61.75)	0.062
	Low	41	68.05(13.769)	63.00 (57.00–82.50)	61.46(13.884)	59.00(51.50–68.00)	0.48
Friendship	Good	35	60.28(15.156)	57.00(48.00–63.00)	58.91(16.220)	55.00(43.00–65.00)	0.087
	Poor	34	67.09(13.527)	63.00(57.00–80.75)	57.04(11.424)	55.50(50.75–67.25)	0.80
Family Relationship	Good	14	61.07(14.143)	53.00(47.00–62.25)	58.98(14.455)	50.00(43.50–63.75)	0.15
	Poor	14	71.36(15.174)	59.00 (57.00–89.25)	62.21(14.110)	64.50(54.75–71.75)	0.62
Job Adjustment	High	20	55.60(10.713)	57.00(49.50–60.75)	53.50(10.511)	54.00(44.75–63.75)	0.20
	Low	8	77.88(18.019)	89.00(59.25–91.50)	71.75(16.334)	68.00(57.50–89.00)	0.36

## Data Availability

Data sharing not available due to restrictions of privacy.

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
