# Peer review of "Effects of Psychotherapy on the Problem Behaviors of Humidifier Disinfectant Survivors: The Role of Individual Characteristics and Adaptive Functioning"

_healthcare, 2023, doi:10.3390/healthcare11152179_

Round 1

Reviewer 1 Report

I read the article very carefully, but I have concerns about the purpose of the research. I did not quite understand the link between harmful effects of humidified disinfectants and psychological variables. I do not understand the utility in demonstrating the effectiveness of non-psychotherapeutic but counseling-type treatment in survivors. I do not feel comfortable accepting this work for publication, it could be an important precedent to support theses that the authors have not well specified

Author Response

Reviewer 1 comment

I did not quite understand the link between harmful effects of humidified disinfectants and psychological variables. I do not understand the utility in demonstrating the effectiveness of non-psychotherapeutic but counseling-type treatment in survivors.

Authors’ Response: We admire your point of view. We would like to elaborate on your opinion. The Korea government and academic community initially focused on the physical symptoms of the damage caused to survivors by humidifier disinfectant. In fact, the chemicals vaporized through the humidifiers and entered the lungs, causing lung tissue damage, fibrosis, and respiratory disease that can be fatal, or even life-threatening, requiring lifelong assistance with oxygen to stay breathing. It also caused catastrophic damage to other organs, leaving survivors with a variety of physical pains, symptoms, and disabilities.

Physical health has been a priority for the government in providing compensation and treatment to survivors. However, in the process of treating survivors, government agencies and academics have come to recognize that these survivors have suffered psychological damage as well as physical damage and are facing significant challenges. It has been reported that when a child is a survivor, the parents are often divorced or come from highly conflicted families, and when a sibling is a survivor, the remaining child often has severe anxiety, depression, and anger management disorders. In many cases, the psychological damage caused by disinfectant is not only permanent, but also severe enough to cause ongoing disruption and difficulty in daily life, so the psychological suffering of family members is also significant. In addition, survivors reported serious psychological symptoms such as anger, helplessness, and frustration as the companies that sold the products and the government battled over the responsibility for compensation. In response, the government has begun to monitor not only the physical but also the psychological health of survivors and to provide mental health services to groups in need.

For psychological support added to somatic symptoms, first, tests were conducted to investigate the mental health status of survivors. Based on the results, mental health services are available not only to the direct survivor, but also to their family members. This study aimed to analyze the psychological status of the survivors through mental health monitoring and services. In particular, we sought to identify differences between groups of survivors who were relatively psychologically healthy and functioning well in the adaptive area and those who reported severe mental distress. These results can help guide the psychotherapy of survivors who are struggling.

It also reveals what factors survivors have that serve as protective factors, which provides a basis for identifying and strengthening the strengths in their personal, environmental, and physical resources. It can also guide the provision of different interventions to survivors who lack these protective factors to find, develop, or strengthen them.

In this study, we found that among survivors, high socioeconomic status, fewer friends, and less socially adjusted groups exhibited more severe psychological problem behaviors. In particular, friendship level was significantly associated with changes in problem behavior from pre- to post-psychotherapy. The group with poor friendships was able to experience positive interpersonal relationships through psychotherapy, such as working alliances and therapeutic relationships with therapist. The group who had poor friendship showed the greatest reduction in problem behavior after psychotherapy.  

In addition, the terms counseling and psychotherapy are often used interchangeably in Korea. Psychotherapy in this study was provided by the certified counseling and clinical psychologists. Treatment in South Korea is primarily provided by clinicians, such as psychiatrists, in hospitals, and in the case of humidifier disinfectant survivors, cases that are clinically indicated for treatment are referred to clinical treatment. Clinical treatment is mainly aimed at controlling symptoms by prescribing medication. Clinical treatment by psychiatrists and psychotherapy by psychologists are often combined. The psychotherapeutic process addresses the survivor's psychological, emotional, and relational difficulties in order to help them adjust, function, and improve their relationships. Therefore, research on the effectiveness of psychotherapy can provide evidence to help improve survivors' psychological well-being and quality of life. We mentioned this point in discussion part.

In text:

The results of this study can be explained by the study of the life adaptation of people who have achieved psychological recovery through individual psychotherapy [34]. Studies on disaster survivors showed that among survivors who suffered psychological difficulties to the extent that they experienced trauma due to disasters [35-42]. Through psychotherapy, individuals achieve psychological recovery by gaining social support, forming a sense of solidarity, and accepting their pain [43-45]. Survivors can learn to cope with stress, enhance life satisfaction, reduce psychological symptoms, function effectively at work and home, and cultivate positive relationships through the psychotherapy process. Finally, it was found that psychological problems were significantly reduced after psychotherapy in the group with poor friendships. This result is in line with the research findings that posit that social support helps psychological recovery after an accident or disaster [46-49]. In addition, this is consistent with the results of studies reporting that happiness and social support are closely related to life satisfaction [50, 51]. Mexico, which is exceptional in the correlation between the average income level, crime rate, and happiness index, has a lower income level and higher crime rate than the United States but a higher happiness index. This index can be attributed to the cultural characteristics of Mexico, where there are stronger family bonds and a higher frequency of contact among family members than in the United States [52]. Individuals with social support are less susceptible to stress and experience fewer psychological difficulties. The group with poor friendships received social support through psychotherapy, which led to a reduction in psychological problems. Summarizing this, it is an indicator that establishing a relational approach to psychological intervention with survivors can be helpful.

The limitations of this study are as follows. Firstly, the subjects were a unique group of survivors of humidifier disinfectant disasters. Therefore, it is necessary to be careful when interpreting the research results for the general public. Secondly, there is a limitation in verifying statistical significance because the number of subjects was relatively small. Although G*Power provided a minimum sample size of 60, it was small to divide the sample into adults and minors and perform all analyses. Thus, it is necessary to replicate these results by increasing the number of subjects in the future. Thirdly, we considered demographic characteristics (e.g., age, SES, compensation presence, etc.) as moderators for treatment effectiveness. However, other potentially critical moderation variables such as the severity of health damages by humidifier disinfectant use, losses of participating family members, the duration of victimization, and the number of psychotherapy sessions should also be included as moderation variables in future studies. Fourthly, this study focused on verifying the effectiveness of a distinctive group of social disaster survivors who immediately needed psychotherapy support. Due to the nature of these subjects, it is ethically problematic to randomly assign survivors to the control group, effectively suspending them from psychotherapy. To explore the effect of psychotherapy in this situation, where a control group could not be secured due to ethical issues, this study observed changes before and after psychotherapy through a time-series design. Because this study has limitations in that it cannot control factors for internal validity from the experimental design, attention should be paid to the interpretation of the results of the study. Lastly, all items were self-reported; therefore, the response could be biased due to faking bad or good (e.g., social desirability). In future studies, the researchers need to utilize objective measures such as significant others’ observations.

Despite all the limitations, this study has meaning: 1) It examined the effect of psychotherapy on survivors of social disasters. 2) Survivors were classified according to differences in their ability to adapt to life even though they experienced the same disaster. The differences in the effectiveness of psychotherapy according to the classified group were examined. 3) It is especially meaningful that we explored which group showed the greatest change due to psychotherapy. Overall, this study provides insights into the group differences among survivors of humidifier disinfectant damage and underscores the positive impact of individual psychotherapy on psychological recovery, particularly in individuals with poor friendship networks.

Reviewer 2 Report

Thank you for the opportunity to review this paper. The aims of the paper are clearly laid out and returned to in the discussion on conclusions. There is a spelling mistake on page 9 line 183 'were is spelt wre instead of were' It is difficult to establish what the response rate was out of the total who took the  courses, perhaps the authors could elaborate upon this. 

Author Response

Thank you for your comments, we revised paper as you suggested. 

Reviewer 2 comment      

1. There is a spelling mistake on page 9 line 183 'were is spelt wre instead of were'.

Authors’ Response: Thank you for your comment. We revised the typo as follow.

In text:

Mann-Whitney tests and Kruskal-Wallis tests were conducted for group differences of each variable at the pre-treatment stage to identify groups that reported more problem behaviors than the other groups.

2. It is difficult to establish what the response rate was out of the total who took the courses, perhaps the authors could elaborate upon this.

Authors’ Response: Thank you for your comment. We have added the response rate in the manuscript.

In text:

This study used data from 69 individuals who survived exposure to toxic humidifier disinfectants and sought individual psychotherapy through a government support program administrated by the National Institute of Environmental Research (NIER) in 2021. Among the survivors aged 13 years and older, a total of 224 individuals received psychotherapy. However, only 69 of them voluntarily participated in the study with informed consent. The response rate among the participants was 30.8%.

Reviewer 3 Report

Dears Authors

General comments

Very interesting and relevant to research and intervention in Korea. Congratulations.

In my opinion, the article has some weaknesses which need to be reflected upon and possibly changed.

However, to make the article even better, I present my reflections according to your article.

 Title

Effects of Psychotherapy on the Problem Behaviors of Humidifier Disinfectant Survivors: The Role of Individual Characteristics and Adaptive Functioning” - Clear and directive

Abstract

- In my view, it needs to be more objective.

Introduction

- Lines 88-90- “This study identified the effectiveness of individual counseling by comparing the  pre-and post-treatment scores of psychological symptoms evaluated by the Achenbach System of Empirically Based Assessment (ASEBA) [23].”- The goal is not clearly identified.

- Lines 88-101- “This study identified the effectiveness of individual counseling by comparing the pre-and post-treatment scores of psychological symptoms evaluated by the Achenbach System of Empirically Based Assessment (ASEBA) [23]. ASEBA provides (…) Additionally, this study examined the moderating role of demographic characteristics in treatment effect to identify the characteristics of survivors who can most benefit from individual counseling.” - I don't see the point of describing the scales in the introduction. It should be revised. It appears between the definition of the objective.

- Line 93-94- “ASEBA measures not only internalizing and externalizing problem behaviors, but also adaptive functioning such as sociality, family relationships, job adjustment, etc.”- I don't think the etc. is appropriate.

- Line 97- “confirmed. (Hong et al., 2022). [25]”- review the points

- Lines 99-101- “Additionally, this study examined the moderating role of demographic characteristics in treatment effect to identify the characteristics of survivors who can most benefit from individual counseling.”- The objective appears cut off by the presentation of the instruments. It should be revised.

- Lines 102-103- “The specific research questions of this study are as follows: First, what are the characteristics of survivors who reported severe psychological symptoms?”- I think the way the question was phrased is not the clearest. And it is justified with a study that presents symptoms “Specifically, as the results of the Ko et al. [18] study, we expect that there are statistical differences in the severity of psychological symptoms by SES.”

- Lines 108-111- “Third, among the survivors, which groups benefited the most from individual counseling? We expect that there are certain characteristics of survivors (e.g., quality of friendship) that moderate the effects of individual counseling on alleviating psychological symptoms.”- Is the expectation supported by any study? If yes, in which one?

- The presentation of research questions and objectives should be revised, in my opinion.

2. Materials and Methods

2.1. Participants

- Lines 116-118 - “1) voluntarily applied for individual counseling through a government support program managed by the National Institute of Environmental Research (NIER) in 2021 and 2) completed pre-and post-measurements.” - No other exclusion criteria were defined, such as previous psychopathological diagnosis? Taking psychotropic drugs? Grief process? Minimum number of sessions attended? It would be important to mention if this screening was done, and if not, to place it in the limitations.

- Lines 102-103- “The age range of the survivors was 13 to 60 years (M = 38.13, SD = 14.75)…” very wide age range - should be evaluated, in my view, by subgroups.

- Line 122- “It should have a therapist who holds a certificate from the Korean Counseling Psychological Association (KCPA) or the Korean Counseling Association (KCA).”- should have or had?

- Lines 124-125- “As of 2021, a total of 42 counseling centers out of  available centers organized by NIER were involved in this study.” - What is the relevance of this information here? This is because earlier you stated that “The survivors received treatment at one of the counseling centers officially approved”.

. Would it be important, from my perspective, to have a control group? Individuals exposed to other situations? Individuals without exposure to stressful events- It would be important to explain and put in the limitations.

2.2 Treatment

- Lines 129-130 - “For controlling therapists’ effects, one or two counseling cases were conducted by one therapist.”- I don't understand what you mean. You should clarify this information.

- Lines 130-132- “The therapists’ theoretical orientations mainly include cognitive-behavioral therapy, psychodynamic therapy, interpersonal psychotherapy, and integrative therapy.”- It would be interesting, in my view, to evaluate this variable as a mediator.

- Lines 132-133- ”The 50-minute sessions were offered unlimited.”- What do you mean? I don't understand what you mean by this statement.

- Lines 133-134 - “The number of sessions ranged from 4 to 41”- Huge variability? How did you control it?

2.3 Measures

2.3.1 Problem behavior scale and 2.3.2 Adaptive functioning scale

- It would be interesting to know the psychometric characteristics of the measured version of the scales.

2.4. Data analysis

- It would be interesting, in my view, to present the demographic and life functioning variables that you are going to study.

3. Results

3.1. Mean differences in problem behavior scores

- Line 183- “Mann-Whitney tests and Kruskal-Wallis tests wre conducted for group”- review typos.

3.2. Treatment effects and interaction with the demographic and life functioning variables

- Lines 214-217- “Because there were some significant differences in problem behaviors before and after treatment according to levels of = friendship, a post-hoc analysis (simple effect comparisons) was conducted. In other words, we examined whether the level of friendship influenced the treatment effects. \ The …”- review typos.

- Line 219- “…ASESBA,…”- review typo.

- Line 222- “problem behaviors dramatically decreased after”- from TMd=55.50 to TMd=63.00 can be considered a dramatic improvement? Important to think about.

4. Discussion

- Lines 242-249 – “This is consistent with a study conducted on Olympic medalists which stated that happiness depends on with whom one compares oneself [33]. According to this study, silver medalists (…) psychological pain with similar SES groups.” - I cannot understand the articulation between the results of the present study and its support with McGraw's results. I could not understand your reasoning process or your support.

- The conclusion needs, in my view, to be further articulated, discussed and supported.

- Lines 326-328- “This result is expected to provide evidence for suggesting a consumer-oriented, precisely customized service in integrated support for survivors of humidifier disinfectants in terms of psychological, physical, and policy aspects.” - How so? In my view, you should be more concrete.

- Lines 328-330- “Additionally, it can be used as valuable empirical data for the psychological support of survivors of other social disasters.” - You have not validated this statement. I think you are making a generalization that has no support in your study.

I wish you good work.

Best regards

Author Response

Thank you for your valuable comments. As you suggested we revised the paper. Please see the attachment that we response your comments one by one. 

Round 2

Reviewer 1 Report

after the review work, this article is a good contribution to the scientific literature.

After a major review, the authors have clarified the controversial points pointed out by reviewer No. 3 in a scientific and rigorous manner. now in my personal view, the work deserves to be published and its scientific value is definitely of a good standard.

Author Response

Reviewer's comment: After the review work, this article is a good contribution to the scientific literature. After a major review, the authors have clarified the controversial points pointed out by reviewer No. 3 in a scientific and rigorous manner. now in my personal view, the work deserves to be published and its scientific value is definitely of a good standard. 

Author's response: Thank you so much for your valuable comments. We really appreciate the reviewers' comments to extend our understanding of the results. 

Reviewer 3 Report

Dears Authors

General comments

Congratulations. I think your work has been significant and the article is now more interesting.

Title: “Effects of Psychotherapy on the Problem Behaviors of Humidifier Disinfectant Survivors: The Role of Individual Characteristics and Adaptive Functioning”- I think it becomes even clearer.

Comments in line with the review:

1.

Comment: To reflect and, in my view, to review:

In the summary you write:

 The results demonstrated significant differences in problems with socioeconomic status (SES), life functioning, friendships, family relationships, and job adjustment in the survivor groups. Groups with high SES, low life functioning, and poor friend relationships had more problem behaviors than other groups. Problem behaviors related to friendship levels were different before and after psychotherapy. After psychotherapy, individuals with limited social connections exhibited a greater decrease in problem behaviors compared to those with strong friendships.

In the conclusion you mention:

Lines 358-374- “In conclusion (…) The results demonstrated the effectiveness of individual psychotherapy in reducing psychological symptoms among survivors of humidifier disinfectant damage. The treatment significantly decreased problem behaviors over time, with medium to large effect sizes. However, no significant interaction effects were found between treatment effects and demographic or life functioning variables, except for friendship.

 8.

Comment: My question is not related to the assumptions but to the support of your expectations/forecasts. “We predict that several traits of survivors (such as the quality of their friendships) will moderate the effectiveness of individual counseling in reducing psychiatric symptoms”. In my view, you should refer to the study(s) that support your expectation/prediction.

10.

Comment: What about taking psychotropic drugs? According to the literature, this variable may influence your results. If you have not taken it into account, I would again urge you to include it in the limitations of the study.

13.

Comment: Thank you for your reply, however, I may not have been clear in my question. For me this information is already given. The question is whether this information is relevant precisely where you have placed it. This is because you had already mentioned “The survivors received treatment at one of the counseling centers officially approved”.

 26.

. “For this purpose, the following analysis was conducted, and the results are as follows:”- this sentence, in my view should be revised. It is the discussion of the study, they should not, in my view, talk again about analysis or results because these have already been presented before. You should present the discussion of the results. “For this purpose, the following results are…:” and discussing them.

. Lines 253-255- “This may be attributed to high-SES individuals comparing their psychological pain with similar SES groups, leading to a higher likelihood of feeling unhappy.”- how do you support this statement? What is your explanation?

. Lines 294-297- “The group with poor friendships received social support through psychotherapy, which led to a reduction in psychological problems. Summarizing this, it is an indicator that establishing a relational approach to psychological intervention with survivors can be helpful”- I have a lot of resistance to this statement. The goal of psychotherapy does not seem to me to be consistent with this goal, nor does the psychotherapist have this role. Is it the result of psychotherapy that has led to the individual's ability to enhance the search for social support? It should be reflected and, in my perspective, you should also review at the end of the discussion when you make the same statement again.

. Lines 322-325- “Practitioners working with survivors of humidifier disinfectant damage should pay attention to individuals from high socioeconomic backgrounds, as they may be prone to comparing their psychological pain with similar socioeconomic groups, potentially leading to greater distress.”- I would like to reiterate my resistance to your interpretation. It lacks, in my view, support.

. Lines- 329-332- “researchers should consider including other critical factors as moderation variables, such as the severity of health damages, losses of family members, duration of victimization, and number of psychotherapy sessions, to better understand their influence on treatment effectiveness.”- what about the issue of medication? Should taking antidepressants/anxiolytics before or during the process not be controlled for? Could it be a mediating variable?

. Line 343- review indentation

. Lines 370-371- “Overall, this research contributes to our understanding” - or for the understanding of the scientific and civil communities?

I wish you a good job.

Author Response

Reviewer 3 comment

  1. In the summary you write:

 “The results demonstrated significant differences in problems with socioeconomic status (SES), life functioning, friendships, family relationships, and job adjustment in the survivor groups. Groups with high SES, low life functioning, and poor friend relationships had more problem behaviors than other groups. Problem behaviors related to friendship levels were different before and after psychotherapy. After psychotherapy, individuals with limited social connections exhibited a greater decrease in problem behaviors compared to those with strong friendships.”

In the conclusion you mention:

Lines 358-374- “In conclusion (…) The results demonstrated the effectiveness of individual psychotherapy in reducing psychological symptoms among survivors of humidifier disinfectant damage. The treatment significantly decreased problem behaviors over time, with medium to large effect sizes. However, no significant interaction effects were found between treatment effects and demographic or life functioning variables, except for friendships.“

Authors’ Response: Thank you for your comment. We clarified our expression.

In text: The results demonstrated the effectiveness of individual psychotherapy in reducing psychological symptoms among survivors of humidifier disinfectant damage. The treatment significantly decreased problem behaviors over time, with medium to large effect sizes. The significant interaction effects were found between treatment effects and friendship levels on problem behaviors.

  1. Comment: My question is not related to the assumptions but to the support of your expectations/forecasts. “We predict that several traits of survivors (such as the quality of their friendships) will moderate the effectiveness of individual counseling in reducing psychiatric symptoms”. In my view, you should refer to the study(s) that support your expectation/prediction.

Authors’ Response: Thank you for your valuable comment. Prior to the research question, we added a preceding study looking at the evidence for it. In the process of psychotherapy, the moderation variables were described.

In text: We predict that several traits of survivors (such as the quality of their friendships) will moderate the effectiveness of individual psychotherapy in reducing psychiatric symptoms. Several scholars [23-24] note that the effectiveness of psychotherapy is a complex interaction of several traits and that what works best for one person may not work the same way for another. Tailoring treatment to the individual's unique characteristics and needs is essential for achieving the best possible outcomes.

  1. Comment: What about taking psychotropic drugs? According to the literature, this variable may influence your results. If you have not taken it into account, I would again urge you to include it in the limitations of the study.

Authors’ Response: Yes, we added psychotropic drugs in the limitation section.

In text: However, other potentially critical moderation variables such as the severity of health damages by humidifier disinfectant use, losses of participating family members, the duration of victimization, taking psychotropic drugs and the number of psychotherapy sessions should also be included as moderation variables in future studies.

  1. Comment: Thank you for your reply, however, I may not have been clear in my question. For me this information is already given. The question is whether this information is relevant precisely where you have placed it. This is because you had already mentioned “The survivors received treatment at one of the counseling centers officially approved”.

Authors’ Response: Thank you for your comment. We agreed with your suggestion that this information is already given. We deleted the sentence.

In text: Out of the 171 psychotherapy centers officially approved by NIER and managed under the government support program in 2021, a total of 42 centers were involved in this study.

  1. “For this purpose, the following analysis was conducted, and the results are as follows:”- this sentence, in my view should be revised. It is the discussion of the study, they should not, in my view, talk again about analysis or results because these have already been presented before. You should present the discussion of the results. “For this purpose, the following results are…:” and discussing them.

Authors’ Response: Thank you for your comment. We changed our expression.

In text: This study aimed to closely examine group differences in the survivors of humidifier disinfectant damage and the effect of individual psychotherapy on the survivor groups. For this purpose, the results are as follows: First…

Lines 253-255- “This may be attributed to high-SES individuals comparing their psychological pain with similar SES groups, leading to a higher likelihood of feeling unhappy.”- how do you support this statement? What is your explanation?

Authors’ Response: Thank you for your comment. We added explanation.

In text: However, survivors from high socioeconomic status (SES) backgrounds reported more problem behaviors compared to those from low- and middle-SES backgrounds. The higher family economic status group's psychological symptoms may be influenced by social comparison. Ko et al. [18] also reported the results that the high-SES group among humidifier disinfectant survivors experienced more psychological problems than the other groups.

Lines 294-297- “The group with poor friendships received social support through psychotherapy, which led to a reduction in psychological problems. Summarizing this, it is an indicator that establishing a relational approach to psychological intervention with survivors can be helpful”- I have a lot of resistance to this statement. The goal of psychotherapy does not seem to me to be consistent with this goal, nor does the psychotherapist have this role. Is it the result of psychotherapy that has led to the individual's ability to enhance the search for social support? It should be reflected and, in my perspective, you should also review at the end of the discussion when you make the same statement again.

Authors’ Response: Thank you for your comment. We changed our expression.

In text: The group with poor friendships may improve their ability to seek social support through psychotherapy, which leads to a reduction in psychological problems. In summary, this indicates that treatment may enhance the survivors' ability to seek social support.

Lines 322-325- “Practitioners working with survivors of humidifier disinfectant damage should pay attention to individuals from high socioeconomic backgrounds, as they may be prone to comparing their psychological pain with similar socioeconomic groups, potentially leading to greater distress.”- I would like to reiterate my resistance to your interpretation. It lacks, in my view, support.

Authors’ Response: Thank you for your comment. We eliminate this sentences.

In text: Based on the findings and limitations of the study, the following recommendations can be made for practice, research, and management. Tailored interventions addressing these specific concerns of survivors may be beneficial.

Lines- 329-332- “researchers should consider including other critical factors as moderation variables, such as the severity of health damages, losses of family members, duration of victimization, and number of psychotherapy sessions, to better understand their influence on treatment effectiveness.”- what about the issue of medication? Should taking antidepressants/anxiolytics before or during the process not be controlled for? Could it be a mediating variable?

Authors’ Response: Thank you for your comment. We changed this part.

In text: In addition to demographic variables, researchers should consider including other critical factors as moderation or mediation variables, such as the severity of health damages, losses of family members, duration of victimization, and number of psychotherapy sessions, to better understand their influence on treatment effectiveness.

Line 343- review indentation

Authors’ Response: Thank you for your comment. We reviewed indentation.

In text: Further research and replication studies are needed to validate the findings and ad-dress the identified limitations. Despite all the limitations, this study has meaning: 1) It examined the effect of psychotherapy on survivors of social disasters. 2) Survivors were classified according to differences in their ability to adapt to life even though they experienced the same disaster.

Lines 370-371- “Overall, this research contributes to our understanding” - or for the understanding of the scientific and civil communities?

Authors’ Response: Thank you for your comment. We changed expression.

In text: Overall, this study contributes to the scientific and civil communities' understanding of the psychological consequences of environmental disasters, underscores the positive impact of individual psychotherapy on survivors' psychological recovery, and highlights the importance of addressing social support and relational approaches in interventions for disaster survivors.
